# Religiosity, Sexual Double Standard, and Intimate Partner Coercive Victimization in Dating Relationships: An Explanatory Model and Psychometric Evidence

**DOI:** 10.3390/bs15030294

**Published:** 2025-03-02

**Authors:** Alhena L. Alfaro-Urquiola, Beatriz Pérez, Francisco Javier Rodríguez-Díaz, Francisco Javier Herrero Diez

**Affiliations:** 1Departamento de Psicología, Universidad de Oviedo, 33009 Oviedo, Spain; uo298006@uniovi.es (A.L.A.-U.); gallego@uniovi.es (F.J.R.-D.); herrero@uniovi.es (F.J.H.D.); 2Departamento de Psicología, Universidad de La Frontera, Temuco 4811322, Chile

**Keywords:** coercive victimization, dating violence, sexual double standard, religiosity, factorial invariance by gender

## Abstract

The literature identifies coercive violence in dating relationships as one of the most prevalent forms of violence and a precursor to more severe types of abuse. While the relationship between the sexual double standard (SDS) and religion with intimate partner violence has been studied, there is a lack of research exploring the direct and indirect influence of religiosity through the promotion of the SDS. This gap is particularly evident when considering measurement invariance by gender, despite these variables being shaped by gender norms. Using a sample of Chilean university students, this study examined the structure and factorial invariance by gender of the Sexual Double Standard Scale (DSS) (*N* = 909) and the Relationship Control Factor Subscale (RCFS) (*N* = 855). Additionally, the study analyzed, for each gender, the impact of religiosity on experiences of coercive victimization within relationships, considering the mediating role of the SDS (*N* = 781). Both instruments revealed different structures by gender, leading to the development of independent models for men (*N* = 278) and women (*N* = 500). Religiosity emerged as a risk factor for victimization in both study groups. However, it also exhibited a protective effect specifically for women.

## 1. Introduction

The study of intimate partner violence (IPV[note 1]), whether during adolescence or early adulthood, has gained increasing importance over time, both in terms of analyzing the phenomenon and intervention ([43]). IPV is understood as “behavior within an intimate relationship that causes physical, sexual, or psychological harm, including acts of physical aggression, sexual coercion, psychological abuse, and controlling behaviors” ([90]) (p. 11). This is linked to adverse outcomes for both physical and psychological health in the short, medium, and long term, which are persistent and clinically significant ([14]).

According to Johnson’s theory ([46], [47]; [49]), IPV manifests in three forms: intimate terrorism, situational couple violence, and violent resistance. The first is driven by control and dominance over the partner, in addition to physical violence, and is more prevalent in clinical samples, associated with controlling behaviors, jealousy, and patriarchal beliefs as risk factors. In contrast, situational violence, more common in community samples, emerges from conflict escalation, with risk factors such as communication difficulties, conflict resolution strategies, and relationship dissatisfaction. Finally, violent resistance is an instinctive response to intimate terrorism attacks by the partner and is largely related to size differences in heterosexual relationships ([56]).

In this context, it is important to consider power dynamics within the couple, where differences in goals, desires, objectives, and beliefs, among other factors, can be sources of conflict, causing one of the members of the relationship to try to influence or control the other according to their interests, which may lead to violence ([4]).

Understanding IPV also requires examining the mechanisms that sustain and perpetuate it. One such mechanism is coercive control, which involves psychological abuse and manipulation aimed at diminishing a partner’s autonomy. This form of control is not always recognized as violence, yet it plays a crucial role in maintaining power imbalances in relationships. Coercive control is a poorly understood dimension of IPV ([55]) and has been found to be associated with more extreme violence, such as sexual violence and even femicide ([41]).

Gender norms, particularly those rooted in the sexual double standard (SDS), further complicate IPV dynamics. The SDS refers to the societal belief that different sexual behaviors are acceptable for men and women, reinforcing power asymmetries in relationships ([8]). According to Kim et al. ([50]), the SDS is related to the recognition of dating violence and sexual assertiveness. Considering that sexuality is experienced differently based on sex, women are often expected to be modest and submissive due to their roles in gestation and breastfeeding, making them solely responsible for their reputation and sexual and reproductive health. In contrast, male sexuality is characterized by strength, power, status, and assertiveness.

The theory of the SDS stands on the belief that boys and men are rewarded and praised for their heterosexual sexual behavior, whereas girls and women are stigmatized for the same ([7]; [52]; [76]). The SDS has been found to correlate with several attitudes and behaviors like attitudes toward condom use ([27]), sexual activity ([89]), sexual concordance ([5]), aggression and men’s violence ([78]), or even rape ([82]). These attitudes not only reinforce gender inequality but also increase risks for women, exposing them to greater vulnerability to sexual coercion, diminished negotiation power in sexual encounters, and overall compromised sexual and reproductive health. For women, moralistic beliefs about their role and sexuality promote a passive sexual role, which may shield them from social stigma but simultaneously place their well-being at risk ([23]).

As with many attitudes and social norms, the SDS can differ between cultures and countries, influencing individuals’ perceptions and experiences of sexuality ([8], [5]). Traditional gender role attitudes are linked to various forms of dating violence ([72]) and violence against women ([42]), highlighting the broader societal implications of these beliefs.

A key factor shaping social and cultural norms related to gender roles and sexuality is religion ([91]), particularly regarding gender roles and expectations in intimate relationships. America is the continent with the largest number of Catholics in the world, with Paraguay being the most Catholic country in the region and Chile one of the countries where Catholicism is most widely expressed in its form of social organization ([75]). This influence has manifested in key areas such as public morality, politics, laws, and education. For example, Chile was the last country in Latin America to legalize divorce in 2004, and civil unions were not legal until 2015 ([54]; [85]). The impact of religion is evident at the structural level and permeates the private sphere, establishing socially and sexually accepted behaviors, such as delaying the onset of sexual relationships in women or promoting virginity as desirable in a partner by men ([39]). Consequently, higher levels of religiosity are associated with stronger adherence to the SDS ([26]; [32]).

At a broader level, religion has been identified as a factor that may shape intimate relationships through the reinforcement of traditional gender norms ([26]; [32]). Some studies indicate that religion can negatively impact how young people socialize and make decisions about their sexuality, reinforcing prejudices ([29]). Additionally, religiosity has been suggested as a factor that may contribute to remaining in abusive relationships due to religious interpretations that reinforce patriarchal norms or limit access to support systems ([87]). However, other research suggests that religiosity does not always have a negative effect on IPV; some studies even report protective effects against certain risk behaviors, including IPV ([29]; [87]).

Findings vary across populations, highlighting the need to contextualize religious influence. For instance, a study with Chilean university students found that religiosity had a protective effect on IPV only at low to moderate levels of religious participation ([54]). Meanwhile, [74] ([74]) found no effect of religiosity on dating victimization among adolescents. However, [24] ([24]) found that Latino men in the U.S. exhibited higher IPV perpetration rates when neither partner was highly religious but held patriarchal beliefs—suggesting that patriarchal ideology, rather than religiosity per se, plays a determining role in IPV dynamics.

### 1.1. Psychometric Properties and Invariant Evaluation of the Sexual Double Standard and Relationship Control Between Sexes

Reliability and validity are essential criteria in the psychometric evaluation of instruments such as the DSS and SRPS, as they guarantee the accuracy and legitimacy in the measurement of psychological constructs. The consistency of an instrument over time and in different contexts is essential to ensure the stability of the measurements ([51]). On the other hand, validity determines the degree to which an instrument effectively measures the phenomenon of interest, and the accuracy of the inferences derived from its scores ([10]). In the field of psychology research, having psychometrically robust scales not only improves the replicability and generalizability of the findings, but also strengthens the empirical basis for the development of evidence-based interventions ([30]).

As mentioned, gender ideology establishes cultural norms that assign different roles and behaviors to men and women. Therefore, it is expected that phenomena influenced by it, such as the SDS and coercive violence in relationships, behave differently for men and women.

Comparing behaviors and attitudes based on sex and gender is very common in research. In fact, many studies differentiate how IPV and associated factors present in men and women ([71]), and how the SDS differs ([26]; [82]). However, these differences are addressed from a sociological perspective and attributed to differences in gender roles and stereotypes ([83]). This strategy does not allow us to determine if the theoretical constructs measured are equivalent between sexes. That is, this strategy does not allow us to evaluate if the measures used, and therefore the theoretical constructs captured with them, are invariant in men and women.

Evaluating the invariance of a scale is essential if comparisons are to be made between groups or over time, since it allows for greater certainty that the differences when evaluating a certain construct are due to variations in the construct itself and not to other factors such as sex, cultural context, or the time at which the evaluation is carried out ([19]).

Authors like [6] ([6]), who have worked with the SDS, recognize the importance of evaluating the factorial and metric invariance of scales to make accurate comparisons between diverse populations. In the case of this variable, sex invariance has been evaluated in various contexts ([9]; [42]). On the other hand, from the strict perspective of psychometrics and invariance, partner violence is a multifactorial phenomenon for which there are various instruments that are not always equivalent ([1]; [59]; [84]), and in many cases, they do not explore invariance ([2]). This leads to unreliable comparisons between genders, either due to measurement issues or different perpetration patterns based on sex ([38]). We agree with O’Hara, Perkins, and Tehhe ([63]) and consider that assessing invariance in scales that evaluate partner violence is essential to determine the causes of differences between men and women.

### 1.2. Objectives and Hypotheses

In summary, we propose three distinct objectives in the present study: (1) to examine the structure and factorial invariance by sex of the Sexual Double Standard Scale (DSS); (2) to examine the structure and factorial invariance by sex of the Relationship Control Factor Subscale (RCFS); and (3) to analyze, differentiated by sex, the association between religiosity and the experience of coercive victimization in the couple (CVC), considering the mediating role of the sexual double standard (SDS).

In line with the proposed objectives, we hypothesize (see Figure 1) that both the DSS (H1) and the RCFS (H2) present different factorial structures for men and women. Additionally, we hypothesize that religiosity is directly associated with the SDS for both men (H3a) and women (H3b) but does not directly impact the experience of CVC for either men (H3c) or women (H3d). In the case of the SDS, we hypothesize that its association with CVC in the men’s group is significant and negative (H3e), while in the women’s group, this association is significant and positive (H3f). Finally, we hypothesize that the SDS plays a mediating role in the relationship between religiosity and the experience of CVC: religiosity is significantly associated with lower coercive victimization for men (H3g), and with higher coercive victimization for women (H3h).

## 2. Materials and Methods

### 2.1. Method

This is a multivariate quantitative study with a non-experimental and cross-sectional design. Structural Equation Modeling (SEM) was used as inferential method ([66]) to address our research objectives, considering probabilistic rather than deterministic causality ([51]).

### 2.2. Participants

The exclusion criteria were as follows: never having been in a romantic relationship and being an adult (29 years or older ([65])). Cases with missing values in the study variables were also excluded. The final sample sizes were as follows:

Total sample target 1: 909 participants (84.16% of cases collected), mean age = 20.69 years (SD = 1.97).

Total sample target 2: 855 participants (79.16% of cases collected), mean age = 20.72 years (SD = 2).

Total sample target 3: 781 participants (72.31% of cases collected), mean age = 20.70 years (SD = 2).

Table 1 presents the sociodemographic characteristics of the full sample and divided by gender. Most participants were enrolled in education, social sciences, and humanities programs, were single, reported a medium family income, identified as non-Indigenous, and described themselves as Catholic or Christian.

### 2.3. Instruments

The battery of instruments was composed of the following scales.

#### 2.3.1. Ad Hoc Sociodemographic Questionnaire

This questionnaire collected participants’ sociodemographic data, including gender and age, as well as descriptive aspects of their romantic relationships (e.g., whether they had ever been in a romantic relationship).

#### 2.3.2. Religious Orientation Scale (ROS; [29], [28])

The ROS, previously validated in Chilean university students ([68]), assesses religiosity through three Likert-scale items: (1) Religious Identification (RI): “To what extent would you say you are religious?” (1 = not at all, 9 = very much); (2) Extrinsic Religiosity (ER): referred to participation in religious events and services, measured by “Other than special occasions (weddings, funerals, baptisms), how often do you attend religious services?” (1 = never, 5 = once a day); and (3) Intrinsic Religiosity (IR): about the experience of sacred connection and its impact on happiness, measured by “Do you consider that your religious beliefs affect your happiness?” (1 = not at all, 9 = very much).

Responses for RI and IR were transformed into a five-point Likert scale, ensuring consistency. The ROS demonstrated adequate internal consistency in the study sample (McDonald’s Omega = 0.871).

#### 2.3.3. Sexual Double Standard Scale (DSS)

The DSS ([17], [16]), adapted for Chilean university students (McDonald’s Omega, 0.918 ([26]), is a unidimensional scale with 10 items rated on a five-point Likert scale (1 = strongly disagree, 5 = strongly agree). It measures the differential evaluation of identical sexual behaviors based on gender with an ordinal α = 0.918 on Chile’s validation and McDonald’s Omega = 0.871 for the present study.

#### 2.3.4. Relationship Control Factor Subscale (RCFS; [70])

##### Instrument Description

The Sexual Relationship Power Scale (SRPS) has two subscales: (a) relationship control, and (b) decision-making dominance ([62]). These unidimensional subscales were part of the scale originally developed in both English and Spanish with a sample of women in the United States, the majority of whom (89%) were Latina (coefficient alpha, 0.84 for English version, 0.88 for Spanish version). The scale has been widely used across various countries, consistently demonstrating robust indicators of validity and reliability in diverse contexts ([70]).

The SRPS full scale consists of 15 Likert-scale items with response options ranging from 1 (strongly disagree) to 4 (strongly agree). This scale has not previously been applied to a Chilean or male sample. It is worth highlighting that the SRPS does not directly assess intimate partner violence (IPV) but rather the distribution of power within relationships. Nevertheless, the “control” factor within the SRPS aligns with constructs measured in other IPV-related scales, such as control and coercion dimensions. The SRPS also aligns theoretically with Johnson’s ([46]) framework on intimate partner violence.

For this study, only the first subscale was used because of its direct link to partner violence, and because in many studies the second one showed very poor psychometric properties. It is worth mentioning that the control subscale has a part specifically linked to condom use ([60]). It is important to highlight those systematic reviews on the SRPS ([48]; [62]) have shown that most studies have been conducted with female samples. Additionally, item wording has been slightly modified, and the number of items has been reduced from the original 15 to either 10 or 13 for the full scale, and to eight items specifically for assessing power dynamics in intimate relationships ([88]).

### 2.4. Procedure

To ensure the clarity of the instructions and instruments, a pilot study was conducted with 30 psychology students in a university classroom during school hours. First, the instrument battery was administered, and then the participants were asked to point out the aspects they found confusing.

After adjustments were made to the instrument battery in accordance with the pilot study, the study sample was collected in university classrooms during school hours. Participants were invited to participate voluntarily and confidentially in classroom settings. Those who agreed signed informed consent forms and completed paper-based surveys. Researchers were present during data collection to address queries. The survey was administered by paper and pencil over a period of six months. The average response time was 45 min, as the data collection was part of another study, which included other scales related to this. There were no incentives for participation. The study was approved by the Scientific Ethics Committee of the Universidad de La Frontera.

### 2.5. Data Analysis

Descriptive statistics (frequencies and central tendency) were used to characterize the sample. To answer Objective 1, a Confirmatory Factor Analysis (CFA) was performed to evaluate the unidimensional structure of the original DSS for Chilean university students. Diagonally Weighted Least Squares (DWLS) estimation was applied to a polychoric correlation matrix, suitable for categorical and ordinal variables ([13]). Fit indices included the Root Mean Square Error of Approximation (RMSEA), Comparative Fit Index (CFI), and Tucker–Lewis Index (TLI), with the following thresholds: for good fit, RMSEA < 0.05, and CFI and TLI > 0.95; and for acceptable fit, RMSEA < 0.08, and CFI and TLI > 0.90 ([51]). Convergent validity was assessed using the Average Variance Extracted (AVE), with AVE ≥ 0.50 indicating adequacy ([35], [37]). Factor loadings ≥ 0.60 were also considered acceptable, given theoretical support ([57]). Additionally, it should be noted that the reliability was evaluated using McDonald’s Omega, internal consistency coefficients which are best suited to ordinal data. Values above 0.7 are considered adequate, and values above 0.9 are considered excellent ([64]). Additionally, we will consider values above 0.65 as acceptable ([25]).

For Objective 2, a cross-validation strategy was employed for the RCFS to assess item discrimination and factorial structure in subsample 1 (*N* = 427) and confirm it in subsample 2 (*N* = 428). Item discrimination was evaluated through skewness (ideal: ±2, acceptable: ±3), kurtosis (ideal: ±7, acceptable: ±10), corrected item–total correlations (>0.3), and initial communalities (>0.3, or tolerable at 0.2–0.3 with adequate model fit and conceptual relevance) ([36]; [51]). An Exploratory Factor Analysis (EFA) using the oblimin rotation method followed the suitability of the correlation matrix (Bartlett’s test and KMO index). A variance explained of ≥40% was deemed acceptable ([35], [37]). CFA was then performed on subsample 2 using the criteria outlined above.

For both the DSS and the RCFS, a multi-group factorial invariance analysis was conducted alongside confirmatory factor analyses (CFAs) to determine whether the latent structure of the instrument is equivalent between men and women. According to Brown ([12]), the analysis begins with configural invariance, which verifies whether the theoretical factor structure is consistent across groups, ensuring that the number of factors and the pattern of factor loadings are equivalent without restrictions. The second level, metric invariance, imposes restrictions on the factor loadings, ensuring that the latent variables are on the same scale. Subsequently, scalar invariance allows for the interpretation of differences in the means of the latent variables between groups, ensuring a shared meaning of the construct in both. Finally, strict invariance incorporates restrictions on residual variances. When this is available and the differences between the different levels do not present changes in the CFI greater than 0.01, RMSEA greater than 0.015, or SRMR greater than 0.030, it ensures that the construct is measured with the same reliability across groups ([77]). This stepwise approach ensures, as far as possible, the equivalence of the constructs measured across different groups and maximizes the validity of intergroup comparisons ([40]).

The confirmation of the absence of factorial invariance required the exploration of the factorial structures of both instruments separately by sex. Due to the small sample size, in the case of men, we limited the analysis to selecting items with adequate discriminative capacity and exploring their structure through Exploratory Factor Analysis (EFA). In the case of women, we used the cross-validation strategy mentioned earlier.

To address Objective 3, the Structural Equation Modeling (SEM) strategy was employed. This approach allowed us to explore not only the direct relationship between religiosity and the sexual double standard on coercive victimization in the couple, but also the mediating role of the sexual double standard in the relationship between religiosity and coercive victimization. This strategy evaluates theoretical, not empirical, causality, and the proposed relationships should be interpreted within this conceptual framework ([51]). It should be noted that after examining the relationship of the study variables with time in the relationship (one year or less/more than one year), religious affiliation (yes/no), and area of study (Education, Social Sciences, and Humanities/others), the latter variable was included in the model as a control variable. Due to the ordinal nature of the data, the robust estimator of the Unweighted Least Squares Mean and Variance adjusted (ULSMV) was used on a polychoric matrix ([13]). The model fit was evaluated using the same fit indices considered in Confirmatory Factor Analysis (CFA).

Descriptive analyses, EFA, McDonald’s Omega, and AVE were conducted in SPSS 24 and JASP 0.18.3.0. CFA and SEM were performed in MPlus 7.

## 3. Results

### 3.1. Structure and Factorial Invariance

#### 3.1.1. Double Sexual Standard by Sex

The multigroup Confirmatory Factor Analysis (Group 1 = male; Group 2 = female) indicates an adequate fit of the study sample (*N* = 906) to the unidimensional structure reported in the literature (X^2^(79) = 238.758, *p* < 0.001; CFI = 0.990; TLI = 0.989; RMSEA = 0.067 (90% CI [0.057, 0.077])). The internal consistency of the scale was adequate for both men (McDonald’s Omega = 0.904) and women (McDonald’s Omega = 0.845). The AVE was also adequate (men AVE = 0.508; women AVE = 0.556). Configurational invariance indicates that the model presents a consistent factorial structure between men and women, with factor variances fixed at 1.0 for both groups (women: estimate = 1.000, 95% CI = [1.000, 1.000]; men: estimate = 1.000, 95% CI = [1.000, 1.000]). This suggests that the number of factors and the basic configuration of item–factor relationships are equivalent between men and women. However, the metric invariance analysis indicates that factor loadings are not equal between groups, as the factor variance estimate for men is slightly lower than that for women (women: estimate = 1.000, 95% CI = [1.000, 1.000]; men: estimate = 0.914, z = 63.553, *p* < 0.001, 95% CI = [0.886, 0.942]). This result suggests that comparisons between groups may not be entirely valid in terms of the relationship between latent and observed variables. Consequently, the factorial structure of the DSS is explored independently for each group.

Descriptive item analyses in the male sample (*N* = 337) indicate adequate skewness and kurtosis values in all cases (see Table 2). Item 3 presents an initial communality of 0.125, and item 8 of 0.146, in addition to a corrected item–total correlation of 0.284. After eliminating both items, the EFA (KMO = 0.903; Bartlett’s test X^2^(28) = 1597.750, *p* < 0.000) reveals a two-dimensional structure that explains 63.1% of the variance. Factor 1, Expectations about Female Sexuality (EFS, items 1, 2, and 9), with factor loadings ranging from 0.532 to 0.598 and sufficient internal consistency (McDonald’s Omega = 0.738), evaluates norms and attitudes that limit female sexuality and penalize sexual behaviors perceived as inappropriate for women. Factor 2, Expectations about Male Sexuality (EMS, items 4, 5, 6, 7, and 10), with factor loadings between 0.513 and 0.728 and adequate internal consistency (McDonald’s Omega = 0.828), measures traditional permissive expectations about male sexuality, including sexual experience, the dominant role, and initiative in sex. Both factors are correlated (r = 0.690, *p* < 0.00).

As the main difference with the male sample, we found that several items in subsample 1 (see Table 2) of the female group (*N* = 285) present negative skewness (items 5, 6, 7, and 10) and leptokurtic kurtosis (6, 7, and 10). This implies low discriminative capacity, so items 6, 7, and 10 were eliminated for this reason. Item 5 was retained because its kurtosis value is within the acceptable range, it has the lowest negative skewness value of the four, and it is the item with the highest correlation level with the set of all items (0.710). Its elimination could mean removing an important contribution to the construct measurement. Additionally, as with the male sample, item 3 presented an extremely low communality (0.103), in addition to a corrected item–total correlation of 0.287. Item 8 also showed a low initial communality (0.161). Both items were eliminated.

Finally, the EFA with subsample 1 (KMO = 0.835; Bartlett’s test X^2^(10) = 774.142, *p* < 0.000) reveals a unidimensional scale of five items with factor loadings ranging from 0.689 to 0.837 (See Table 2), explaining 60.9% of the variance. Its internal consistency is sufficient (McDonald’s Omega = 0.787). The CFA in subsample 2 (*N* = 284, X^2^(5) = 38.777, *p* < 0.001; CFI = 0.995; TLI = 0.991; RMSEA = 0.052 (90% CI [0.000, 0.107])) showed an excellent fit to the data. The AVE is above 50% (AVE = 0.574). The internal consistency of this structure is adequate in both subsample 1 (McDonald’s Omega = 0.787) and subsample 2 (McDonald’s Omega = 0.809).

#### 3.1.2. Relationship Control Factor Subscale by Sex

In subsample 1 (*N* = 427), no item presents unacceptable levels of skewness or kurtosis. The initial communalities of the items exceed 0.2 in all cases, and the corrected item–total scale correlations exceed 0.3 in all cases. Consequently, all items are retained as part of the instrument.

The EFA (KMO = 0.902; Bartlett’s test X^2^(105) = 5151.145, *p* < 0.001) reveals a structure of 15 items and four factors: (a) Perceived Relational Submission (PRS, items 5, 7, 11, 12, and 14). This factor evaluates the subjective perception of imbalance, discomfort, or discontent, characterized by feelings of inequality in commitment, benefit, or dedication between the members of the relationship (McDonald’s Omega = 0.798). (b) Imposition of the Couple (IC, items 3, 6, 9, and 10). This factor measures the imposition of the partner’s desires and decisions over one’s own (McDonald’s Omega = 0.733). (c) Restrictive Relational Control (RRC, items 8, 13, and 15). This factor evaluates the active role of the partner in specific relational control behaviors (McDonald’s Omega = 0.638). (d) Male Coercion in Negotiating Safe Sex (MCSS, items 1, 2, and 4). This factor refers to the control exerted by the male partner in decisions about condom use, characterized by reactions of hostility, violence, or accusations that hinder the negotiation of safe practices in the relationship (McDonald’s Omega = 0.806). Together, these factors explain 67.1% of the variance (see Appendix A).

This model in subsample 2 (*N* = 428) showed an adequate fit (X^2^(84) = 190.451, *p* < 0.001; CFI = 0.979; TLI = 0.974; RMSEA = 0.055 (90% CI [0.044, 0.065])). The internal consistency was adequate for the PRS factor (McDonald’s Omega = 0.778); IC factor (McDonald’s Omega = 0.789); and MCSS factor (McDonald’s Omega = 0.824). As in subsample 1, the RRC factor showed poor internal consistency (McDonald’s Omega = 0.646). Finally, it should be noted that the AVE exceeded 0.5 in all cases (PRS factor, AVE = 0.562; IC factor, AVE = 0.646; RRC factor, AVE = 0.584; MCSS factor, AVE = 0.824). Again, the questionnaire structure presents configurational invariance between sexes (women, factors 1, 2, 3, and 4: estimate = 1.000, 95% CI = [1.000, 1.000]; men: estimate = 1.000, 95% CI = [1.000, 1.000]), but not metric invariance (women, factors 1, 2, 3, and 4: estimate = 1.000, 95% CI = [1.000, 1.000]; men: (1) estimate = 0.937, z = 5.364, *p* < 0.001, 95% CI = [0.595, 1.280]; (2) estimate = 1.150, z = 5.065, *p* < 0.001, 95% CI = [0.705, 1.594]; (3) estimate = 1.192, z = 4.486, *p* < 0.001, 95% CI = [0.671, 1.713]; (4) estimate = 1.348, z = 4.658, *p* < 0.001, 95% CI = [0.781, 1.915]).

Examining the structure of the RCFS in the male sample (*N* = 308) involves questioning the adequacy of some items. Although all items met the criteria ensuring adequate discriminative capacity (see Table 3), items 1, 2, and 4 were eliminated based on theoretical criteria. These items were originally formulated to assess male coercion over women in negotiating safe sex. Therefore, they cannot be applied to the male sample.

During the EFA, we identified issues with the performance of item 5, “When my partner and I are together, I tend to be rather quiet”. It presented an anomalous factor loading, greater than 1; the uniqueness was negative; and it did not group with other items in any factor. This indicates that the item does not significantly contribute to the model, nor is it supported by consistent groupings with other items. For these reasons, item 5 was eliminated. The final EFA (KMO = 0.883; Bartlett’s test X^2^(55) = 1692.864, *p* < 0.001) revealed a scale of 11 items and three factors that explains 56.6% of the variance: factors 1, 2, and 3 mentioned earlier are retained, with the difference that item 6 migrates from factor 2 to factor 1 (see Table 3). The internal consistency of the PRS factor (McDonald’s Omega = 0.749) and the IC factor (McDonald’s Omega = 0.706) are adequate. The internal consistency of the RRC factor was acceptable (McDonald’s Omega = 0.667).

Finally, we performed cross-validity analysis on the female sample. The scale items in subsample 1 (*N* = 274) show adequate discriminative capacity (see Table 4). The EFA (KMO = 0.806; Bartlett’s test X^2^(105) = 3815.486, *p* < 0.001) revealed a structure of 15 items and four factors, explaining 69.6% of the variance. The organization of items by factors is consistent with that found in the EFA with the general subsample 1, except for item 5, which migrates from the PRS factor to the IC factor. Additionally, the internal consistency was adequate for the PRS factor (McDonald’s Omega = 0.784); IC factor (McDonald’s Omega = 0.786); and MCSS factor (McDonald’s Omega = 0.788), while the internal consistency of the RRC factor is deficient (McDonald’s Omega = 0.638). This structure fit in subsample 2 of women (X^2^(84) = 110.142, *p* < 0.001; CFI = 0.988; TLI = 0.985; RMSEA = 0.034 (90% CI [0.011, 0.050])). Again, the PRS factor (McDonald’s Omega = 0.756); IC factor (McDonald’s Omega = 0.721); and MCSS factor (McDonald’s Omega = 0.759) showed adequate internal consistency, while the RRC factor (McDonald’s Omega = 0.579) obtained deficient internal consistency. The AVE only exceeded 0.5 in factor 4 (PRS factor, AVE = 0.443; IC factor, AVE = 0.354; RRC factor, AVE = 0.314; MCSS factor, AVE = 0.547).

### 3.2. Impact of Religiosity on the Experience of Coercive Victimization in the Couple, Considering the Mediating Role of the Sexual Double Standard

#### 3.2.1. Men’s Model

The model proposed to address Objective 3 (Figure 1) is analyzed with the male sample (*N* = 278), considering the modifications resulting from the factorial structure of the DSS and RCFS in men (Figure 2). The variable religious affiliation was discarded as a control variable for the religiosity variable, as its use implies a significant loss of sample size, which in turn implies a lack of convergence of the model. The variable time in relationship was used as a control variable for the RFL factors. However, it was discarded due to the lack of statistically significant results (PRS, β = −0.054, SE = 0.071, *p* = 0.449; IC, β = 0.063, SE = 0.074, *p* = 0.397; RRC, β = −0.020, SE = 0.079, *p* = 0.797). The variable Study Area is incorporated as a control variable over the religiosity variable.

After adding the correlation between the DSS factors (r = 0.861, *p* < 0.001), this model showed an adequate fit (X^2^(215) = 270.456, *p* < 0.001; CFI = 0.938; TLI = 0.927; RMSEA = 0.030 (90% CI [0.017, 0.041])). The factor loadings of the indicators in their respective latent variables were as expected and of adequate magnitude (all greater than 0.475). The religiosity variable showed a direct positive effect on EFS (β = 0.290, SE = 0.063, *p* < 0.001) and EMS (β = 0.272, SE = 0.062, *p* < 0.001), but no direct effect on the RCF factors. Additionally, we found that students in the Education, Social Sciences, and Humanities area have lower levels of religiosity (β = −0.241, SE = 0.061, *p* < 0.001). On the other hand, while EFS has no effect on the RCF factors, EMS showed a significant and positive direct effect on PRS (β = 0.586, SE = 0.201, *p* = 0.003) and RRC (β = 0.585, SE = 0.192, *p* = 0.002). It showed no direct effect on IC.

An indirect effect of religiosity on the RCF factors through EFM is ruled out, and the indirect and positive effect of religiosity through the mediating variable EMS on PRS (β = 0.119, SE = 0.038, *p* = 0.002) and RRC (β = 0.160, SE = 0.040, *p* < 0.001) is confirmed. Higher religiosity leads to higher PRS and RRC through the mediating variable EMS. This model explains 21.2% of the variance of PRS and 35.4% of the variance of RRC.

#### 3.2.2. Women’s Model

The model proposed to address Objective 3 (Figure 1) is analyzed with the female sample (*N* = 500), considering the modifications resulting from the factorial structure of the DSS and RCFS in women (Figure 3). The variable religious affiliation was discarded as a control variable for the religiosity variable, as its use implies a significant loss of sample size. The variable time in relationship was used as a control variable for the RFL factors. However, it was discarded due to the lack of statistically significant results (PRS, β = −0.037, SE = 0.057, *p* = 0.514; IC, β = 0.057, SE = 0.056, *p* = 0.305; MCSS, β = 0.055, SE = 0.069, *p* = 0.427). The variable Study Area is incorporated as a control variable over the religiosity variable. It is worth mentioning that, although the factorial structure of the RCFS reveals four factors, we decided not to include the RRC factor in the model due to its deficient values in internal consistency and AVE. The PRS and IC factors also showed an AVE below 0.5 in subsample 2. However, the decision was made to retain both factors in the model due to their adequate internal consistency (>0.70) and the high and consistent factor loadings of their items.

This model showed an adequate fit (X^2^(179) = 212.955, *p* = 0.0421; CFI = 0.981; TLI = 0.977; RMSEA = 0.019 (90% CI [0.004, 0.029]). The factor loadings of the indicators in their respective latent variables were as expected and of adequate magnitude (all greater than 0.695). Religiosity positively impacts the SDS (β = 0.263, SE = 0.055, *p* < 0.001) and negatively impacts PRS (β = −0.130, SE = 0.062, *p* = 0.036) and MCSS (β = −0.247, SE = 0.080, *p* = 0.002). It has no impact on IC. Additionally, we again found that students in the Education, Social Sciences, and Humanities area have lower levels of religiosity (β = −0.257, SE = 0.044, *p* < 0.001). The SDS had a direct and positive impact on PRS (β = 0.319, SE = 0.063, *p* < 0.001), IC (β = 0.384, SE = 0.060, *p* < 0.001), and MCSS (β = 0.487, SE = 0.065, *p* < 0.001).

Finally, religiosity shows an indirect and positive effect on PRS (β = 0.084, SE = 0.023, *p* < 0.001), IC (β = 0.101, SE = 0.026, *p* < 0.001), and MCSS (β = 0.128, SE = 0.033, *p* < 0.001). That is, higher religiosity leads to higher PRS, RRC, and MCSS through the mediating variable SDS. This model explains 9.7% of the variance of PRS, 13.8% of the variance of IC, and 23.5% of the variance of MCSS.

## 4. Discussion

The findings of this study reveal gender differences in the structures of the DSS (Double Standard Scale) and the RCFS (Relationship Control Factor Subscale), which do not align with the previous literature. The Structural Equation Modeling (SEM) results indicate that, for men, religiosity is associated with CVC (coercive victimization in the couple) through an indirect positive effect via the SDS. However, for women, a direct negative effect and an indirect positive effect are observed.

The factorial structure of the DSS shows significant differences according to gender, suggesting variability in the theoretical construct that the scale evaluates. In both samples, items 3 and 8 were removed due to their low contribution to measuring the construct, a finding consistent with psychometric studies conducted in El Salvador ([80]), Spain ([86]), Brazil ([79]), and Peru ([81]). In the female version, items 6, 7, and 10 were also eliminated due to their low discriminative capacity; coincidentally, these three items focus on men, while four of the five items retained in the female scale (items 1, 2, 5, and 6) focus on women. The SDS exhibited a unidimensional structure in the female sample, whereas two factors were identified in the male sample, “Expectations about Female Sexuality” and “Expectations about Male Sexuality”, which are correlated.

The results highlight that beliefs about normative sexual roles vary between genders: women tend to reject statements centered on masculine roles, while normative beliefs regarding female sexual behavior are accepted by both genders. This finding suggests that traditional male and female expectations do not manifest with equal intensity or homogeneity. Previous instruments such as the Spanish version of the Sexual Double Standard Scale ([8]; [33]) also allow for distinguishing differentiated profiles of the SDS. However, until now, the unifactorial structure has been maintained in prior cultural adaptations.

The differences in responses between men and women can be explained by factors such as gender socialization ([11]) and the increasing participation of women in educational and labor contexts, which facilitates their exposure to progressive ideologies. In contrast, men progress more slowly due to adherence to traditional roles that privilege them ([69]; [73]). However, these differences remain anchored in roles of male sexual domination.

The theory of ambivalent sexism ([31]) helps to understand this duality: while women reject hostile sexism, they may adhere to benevolent sexism, reinforcing roles of dependence and passivity ([18]; [22]). In this regard, [33] ([33]) demonstrate how women tend to exhibit more critical attitudes towards male sexual freedom but may more readily accept expectations that perpetuate “female sexual shyness”. This pattern reflects an internalization of traditional social norms that assign women a more reserved or passive role in sexuality while rejecting standards that favor male freedom.

The factorial structure of the SRPS (Sexual Relationship Power Scale) differs significantly between sexes, supporting the second hypothesis of the study. In the female sample, all items from the original scale were retained, while in the male sample items, 1, 2, 4, and 5 were removed due to poor fit. Similar findings were reported in the South African version of the 13-item scale ([44]), where after factor analysis eight items remained for women and nine for men, sharing six common items. The removal of items related to condom use in previous studies ([21]) may be attributed to cultural sensitivities or gender biases in interpreting these behaviors. Although structural differences are not extensive, it is notable that the MCSS (Male Control Sexual Scale) factor was suppressed in men and there was divergence in the RCFS (Relationship Control Factor Subscale). These results suggest that perceptions of relational control and power dynamics differ between genders, which should be further examined in future studies with a cultural perspective.

In response to objective 3, two structural equation models were developed, one for the male sample and one for the female sample. The male model confirms Hypotheses 3a and 3c but refutes Hypotheses 3e and 3g. On the other hand, the female model validates Hypotheses 3b, 3f, and 3h, although it also refutes Hypothesis 3d.

Regarding the relationship between the sexual double standard (SDS) and the experience of coercive victimization in the couple (CVC), both matches and discrepancies with the proposed hypotheses were found. In the case of the male sample, the impact of the SDS on the CVC is complex, since while the factor “Expectations about Female Sexuality” (EFS) does not show a direct effect on the CVC, the factor “Expectations about Male Sexuality” (EMS) does present a significant effect. This finding suggests that traditional male expectations regarding sexuality, which position men in power roles (EMS), are not associated with the perception of domination or control in decision-making by the partner (Partner Imposition—PI). However, such expectations can contribute to relational distress (Perceived Relational Stress—PRS) and controlling behaviors within relationships, negatively affecting men (Relational Control Restriction—RCR). Although unexpected, this result aligns with the literature indicating that exaggerated relationship expectations can lead to dissatisfaction ([15]).

According to the Social Distance Theory of Power ([58]), power imbalances within relationships can alter perceptions of partner behavior, ultimately affecting relational satisfaction ([45]). Men adhering to traditional gender roles may interpret their partner’s actions negatively when these deviate from their expectations, potentially creating a sense of imbalance and discomfort within the relationship. In some cases, this dynamic could lead to a perception of controlling behavior by the partner.

For the female sample, the impact of the SDS on CVC was also significant and positive, though its interpretation differs from that of men. In this case, adherence to submissive sexual beliefs associated with the SDS may contribute to engaging in relationships with unbalanced power dynamics, where male partners assume a dominant role (Imposition of the couple). Women in these relationships experience relational distress, often linked to unequal commitment levels (Perceived Relational Submission), and are exposed to unsafe sexual practices (Male Coercion in Negotiating Safe Sex). This finding is consistent with a review of 22 studies by [6] ([6]), which documented how restrictive attitudes toward female sexuality negatively impact sexual well-being and increase vulnerability to risky sexual behaviors.

As anticipated, religiosity was positively associated with the SDS in both men and women, consistent with previous studies linking religiosity to traditional gender norms and the sexual double standard ([3]; [20]; [26]; [32]; [39]; [53]; [54]; [85]). However, no direct effect of religiosity on CVC was observed in men or on the IC factor in women ([24]; [74]). Instead, religiosity exhibited a significant and negative association with the PRS and MCSS factors in women, suggesting that religiosity serves as a protective factor or a buffer against relational dissatisfaction and male control in safe sex negotiations. This phenomenon has been partially documented by [54] ([54]) and [24] ([24]), who argued that in moderate religious contexts, religiosity can reduce relationship violence, provided it is not rooted in patriarchal ideologies. Similarly, other studies (e.g., [3]) have indicated that religiosity can empower women to set boundaries in their sexual relationships, reducing exposure to multiple partners and risky behaviors ([61]; [67]).

Finally, although no direct effect of religiosity on CVR was identified in men and only a positive direct effect was observed in women, religiosity was indirectly associated with CVR through the SDS in both samples. This indirect effect supports Hypothesis 3h for women, while contradicting Hypothesis 3g for men. Consequently, the promotion of traditional sexual gender norms by may represent a risk factor for CVR, regardless of gender. This finding aligns with [54] ([54]), who noted that the relationship between religiosity and victimization in romantic relationships is non-linear. While low to moderate levels of religiosity may be protective, extreme levels can increase risk, particularly when discrepancies exist in the religiosity levels between partners ([24]).

This study has a number of limitations. First, the results may be biased by the sample, which consists entirely of university students, mostly from the social sciences and humanities. In addition, although religious participants predominate, evangelicals are underrepresented. Approximately half of the participants were not in a romantic relationship at the time they took part in the study, which may have influenced their recall when responding.

Secondly, the RCFS was designed for women and the RRC factor showed poor internal consistency in this sample, so it could not be used in the explanatory model. Further-more, the DSS measures perceptions of social support for the SDS, whereas individual attitudes are best reflected by implicit measures and intrasubject designs ([27]). This has implications for the interpretation of constructs and findings. Finally, other variables associated with coercive violence in the literature could have been included in the construction of the models.

Given that this is an inferential study based on the relationships between variables rather than experimentation, and considering that the study was conducted with a community sample—where high rates of intimate partner violence are generally not observed—it is not possible to establish causal relationships.

One key methodological limitation of this study pertains to measurement invariance. While we considered the possibility of testing for partial invariance, the challenges associated with this approach—particularly the limited data available for men and the lack of a well-established theoretical framework for the constructs across genders—led us to conduct separate structural analyses by gender. Future research could explore partial invariance in larger and more diverse samples, allowing for a deep analysis which would enable a more nuanced understanding of the common and unique elements of these constructs across genders.

Although the inclusion of control variables in the structural equation model (SEM) aimed to reduce potential confounding effects, unmeasured sociocultural and individual factors may have influenced the results. Given that SEM assesses theoretical rather than empirical causality, the proposed relationships should be interpreted within the study’s conceptual framework rather than as definitive causal evidence.

As Greenland and Pearce ([34]) highlight, SEM findings should be approached with caution, particularly in complex relationships where unaccounted variables may still exert an influence. Future research should incorporate longitudinal designs, probability sampling, and a broader set of control variables to enhance external validity and further refine the theoretical model.

## 5. Conclusions

In the data from this study, the “Sexual Double Standard Scale” exhibits distinct factorial structures for each gender. While the male version comprises two factors related to traditional feminine and masculine sexual roles, the female version follows a unidimensional structure, measuring support for feminine sexual roles. These findings align with previous research by [7] ([7], [8]) on differential gender support for sexual double standard typologies, suggesting that differences in support for the sexual double standard between men and women are not merely quantitative but also qualitative.

Similarly, the “Relationship Control Factor”, a subscale of the “Sexual Relationship Power Scale”, reveals a different factorial structure for men and women ([1]), indicating that perceptions of relationship control and power dynamics vary across genders. In essence, both constructs hold different meanings for men and women. These findings underscore the importance of assessing factorial invariance between genders when measuring constructs that are strongly shaped by gender norms.

Moreover, the explanatory models analyzed suggest that religiosity has both direct and indirect effects on coercive victimization. In women, religiosity directly reduces coercive victimization but also has an indirect positive association with it. In men, however, the effect is exclusively direct and positive. Thus, while religiosity may function as a protective factor for women, it simultaneously appears to be a risk for both genders by reinforcing the sexual double standard. Furthermore, theoretical differences in constructs must be taken into account, as they influence the interpretation of explanatory models.

For men, religiosity is associated with greater perceived relational submission and restrictive relational control, reinforcing expectations regarding male (but not female) sexuality. This unexpected finding suggests that adherence to the sexual double standard can also result in male victimization, despite its role in reinforcing male sexual power. Meanwhile, in women, higher religiosity is linked to lower perceived relational submission and reduced male coercion in negotiation. However, it is also associated with higher scores on both scales and with the reinforcement of relationship dynamics through the sexual double standard. These results highlight the complexity of the relationship between religiosity and female victimization, in which the sexual double standard plays a significant role.

Ultimately, for both men and women, the sexual double standard emerges as a critical variable to consider in intervention strategies, with the goal of reducing the negative effects of religious adherence on coercive victimization.

## Figures and Tables

**Figure 1 behavsci-15-00294-f001:**
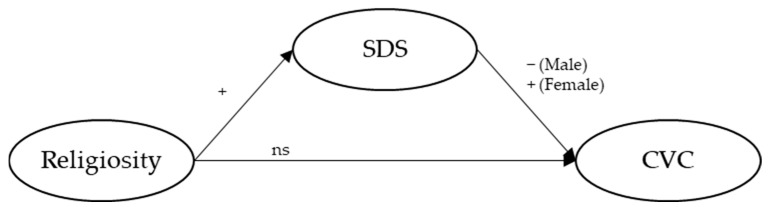
Proposed structural equation model.

**Figure 2 behavsci-15-00294-f002:**
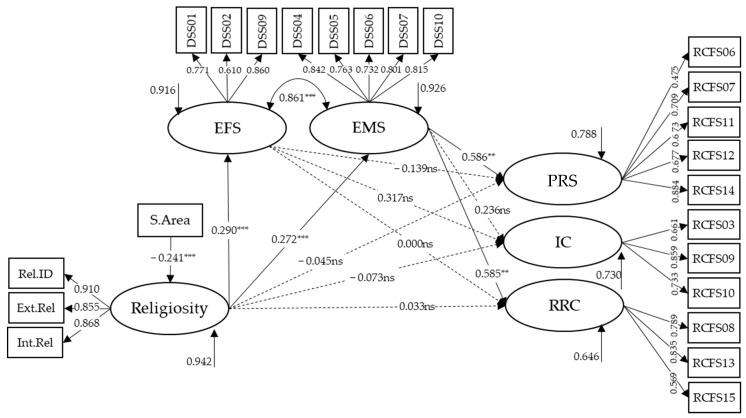
Standardized coefficients of the male model with control variables. Rel.ID = Religious Identification; Ext.Rel = Extrinsic Religiosity; Int.Rel = Intrinsic Religiosity; DSS = Double Standard Scale; EFS = Expectations about Female Sexuality; EMS = Expectations about Male Sexuality; RCFS = Relationship Control Factor Subscale; PRS = Perceived Relational Submission; IC = Imposition of the Couple; RRC = Restrictive Relational Control; S.Area = Study Area; ** *p* < 0.01; *** *p* < 0.001.

**Figure 3 behavsci-15-00294-f003:**
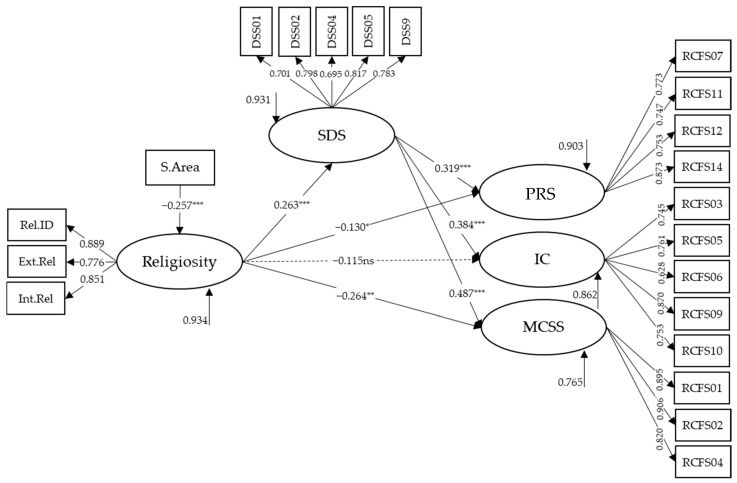
Standardized coefficients of the female model with control variables. Rel.ID = Religious Identification; Ext.Rel = Extrinsic Religiosity; Int.Rel = Intrinsic Religiosity; DSS = Double Standard Scale; SDS = Sexual Double Standard; RCFS = Relationship Control Factor Subscale; PRS = Perceived Relational Submission; IC = Imposition of the Couple; MCSS = Male Coercion in Negotiating Safe Sex; S.Area = Study Area; * *p* < 0.05; ** *p* < 0.01; *** *p* < 0.001.

**Table 1 behavsci-15-00294-t001:** Descriptive data in the total sample and by gender.

		Target Sample 1 (*N* = 909)			Target Sample 2 (*N* = 855)			Target Sample 3 (*N* = 781)		
		T ^1^. *n* (%)	M. *n* (%)	F. *n* (%)	T ^1^. *n* (%)	M. *n* (%)	F. *n* (%)	T ^1^. *n* (%)	M. *n* (%)	F. *n* (%)
Sex	Male (M)	377 (37.2)			305 (35.8)			278 (35.7)		
	Female (F)	569 (62.8)			547 (64)			500 (64.3)		
Study Area	Education, Social Sciences, Humanities	424 (46.7)	133 (39.5)	291 (51.2)	398 (46.7)	122 (40)	276 (50.5)	360 (46.2)	111 (39.9)	249 (49.9)
	Others (Medicine, Odontology, Engineering and Sciences, Agricultural and Forestry Sciences, and Legal and Business Sciences)	483 (53.3)	204 (60.3)	273 (48.7)	455 (53.3)	183 (60)	270 (35.1)	429 (53.8)	167 (60)	251 (50.1)
Family income	Low (−$250,000)	115 (14.4)	46 (15.2)	69 (13.8)	108 (14.5)	42 (15.6)	66 (13.9)	99 (14.2)	38 (15.3)	61 (13.9)
	Medium ($250,000–$1,000,000)	501 (62.5)	184 (60.9)	317 (63.5)	470 (63)	167 (61.9)	303 (63.7)	434 (55.6)	156 (62.7)	278 (63.3)
	High (+$1,000,000)	185 (23.1)	72 (23.8)	113 (22.6)	168 (22.5)	61 (22.6)	107 (22.5)	155 (22.5)	55 (22.1)	100 (22.8)
Marital status	Single (currently in a romantic relationship)	481 (53.1)	159 (47.2)	321 (56.6)	472 (55.3)	151 (48.5)	319 (58.5)	428 (54.8)	136 (48.9)	290 (58.1)
	Single (past romantic relationship)	413 (45.6)	172 (51)	240 (42.3)	369 (43.2)	148 (48.5)	220 (40.4)	342 (43.8)	138 (49.6)	203 (40.7)
	Married/Split	12 (1.3)	6 (1.8)	6 (1.1)	12 (1.4)	6 (2)	6 (1.1)	10 (1.3)	4 (1.5)	6 (1.2)
Indigenous ethnicity	No	796 (87.6)	292 (86.7)	501 (88)	745 (87.1)	262 (85.9)	480 (87.8)	679 (87)	239 (86)	437 (87.4)
	Mapuche	104 (11.4)	41 (12.2)	63 (11.1)	101 (12)	39 (12.9)	62 (11.2)	93 (11.9)	35 (12.6)	58 (11.6)
	Other	9 (1)	4 (1.1)	5 (1)	9 (0.9)	4 (1.2)	5 (1)	9 (1.1)	4 (1.4)	5 (1)
Religion	Catholic/Christian	342 (49.1)	107 (43.3)	235 (52.5)	322 (49.2)	95 (42.8)	227 (52.8)	304 (49.4)	87 (39.3)	217 (53.8)
	Evangelic/Protestant	81 (11.6)	28 (11.3)	135 (11.8)	78 (11.9)	27 (12.2)	51 (11.9)	74 (12)	26 (12.3)	48 (11.9)
	Other	44 (6.3)	19 (7.7)	25 (5.6)	40 (6.1)	16 (7.2)	24 (5.6)	38 (6.2)	19 (7.4)	26 (5.7)
	None/agnostic/atheist	230 (33)	93 (37.7)	135 (30.1)	214 (32.7)	84 (37.8)	128 (29.8)	200 (32.5)	83 (39.3)	115 (28.5)

^1^ T = Total.

**Table 2 behavsci-15-00294-t002:** Descriptive statistics of DSS items and factor weights in a sample of males and females.

	Males								Females							
	M	SD	Skew	Kurt.	ITC	ICom	Fct Weights		M	DS	Skew	Kurt.	CIT	ICom	Fct Weights	
							EFS	EMS							Ss1	Ss2
1. It is expected that a woman will be less sexually experienced than her partner	2.02	1.012	0.803	0.007	0.576	0.440	0.731		1.68	0.935	1.388	1.338	0.549	0.406	0.742	0.776
2. A woman who is sexually active is less likely to be desired as a partner	2.39	1.175	0.443	−0.855	0.488	0.328	0.761		1.67	0.991	1.415	1.263	0.578	0.448	0.810	0.804
3. A woman should never appear to be prepared for a sexual encounter	2.32	1.239	0.752	−0.368	0.327	0.128	--	--	2.07	1.409	1.079	−0.229	0.287	0.103	--	--
4. It is important for men to be experienced sexually to be able to teach the woman	1.72	0.823	1.169	1.356	0.623	0.481		0.699	1.35	0.674	2.365	6.92	0.479	0.321	0.689	0.748
5. A “nice” woman has never had a one-night stand, but a man is expected to have had one	1.63	0.887	1.538	2.282	0.617	0.412		0.513	1.31	0.753	3.017	9.68	0.632	0.535	0.837	0.754
6. It is important for a man to have multiple sexual encounters to gain experience	1.69	0.904	1.441	1.93	0.573	0.492		0.961	1.19	0.53	3.876	19.812	0.556	0.456	--	--
7. In sex, the man must take the dominant role and the woman the passive role	1.81	0.947	1.037	0.507	0.618	0.472		0.728	1.23	0.584	3.02	10.59	0.575	0.46	--	--
8. It is acceptable for a woman to carry her own condoms	1.71	0.974	1.535	2.175	0.284	0.178	--	--	1.48	0.902	2.445	6.132	0.304	0.161	--	--
9. It is worse for a woman to be promiscuous than for a man	1.92	1.055	0.981	0.146	0.689	0.512	0.501		1.44	0.877	2.072	3.655	0.639	0.487	0.815	0.686
10. It is the man’s decision to initiate sex	1.56	0.77	1.371	1.878	0.594	0.410		0.666	1.24	0.633	3.534	14.828	0.554	0.445	--	--

M = media; SD = standard deviation; Skew = skewness; Kurt. = kurtosis; ITC = item-total correlation; ICom = initial communalities; EFS = Expectations about Female Sexuality; EMS = Expectations about Male Sexuality; Ss = subsample; -- = items removed from the scale.

**Table 3 behavsci-15-00294-t003:** Descriptive data of the SRPS items and factorial weights in the male sample.

		M	SD	Skew	Kurt.	ITC	ICom	Fct. Weights		
								PRS	IC	RRC
1. If I asked my partner to use a condom, she would become violent.		1.35	0.547	1.272	0.654	0.593	0.769	--	--	--
2. If I asked my partner for a condom, she would become furious.		1.32	0.547	1.437	1.125	0.609	0.764	--	--	--
3. Most of the time, we do what my partner wants to do.		1.88	0.716	0.176	−1.039	0.49	0.308		0.371	
4. If I asked my partner to use a condom, he/she would think I am having sex with other people.		1.52	0.696	0.967	−0.348	0.504	0.455	--	--	--
5. When my partner and I are together, I tend to be rather quiet.		1.67	0.702	0.550	−0.845	0.493	0.321	--	--	--
6. My partner does what he/she wants, even if I don’t want them to.		1.88	0.767	0.197	−1.279	0.399	0.200	0.467		
7. I feel trapped or confined in our relationship.		1.70	0.736	0.522	−1.003	0.606	0.405	0.609		
8. My partner doesn’t let me wear certain types of clothes.		1.40	0.588	1.143	0.306	0.571	0.420			0.362
9. My partner has more influence than I do in important decisions that affect us.		1.58	0.657	0.688	−0.573	0.627	0.488		0.620	
10. When my partner and I disagree, my partner almost always gets her/his way.		1.94	0.780	0.091	−1.352	0.52	0.397		0.819	
11. I am more committed to the relationship than my partner is.		1.91	0.723	0.134	−1.076	0.5	0.327	0.594		
12. My partner might be having sex with someone else.		1.45	0.641	1.115	0.111	0.542	0.402	0.746		
13. My partner tells me who I can spend my time with.		1.50	0.672	0.977	−0.244	0.613	0.444			0.780
14. Overall, my partner benefits more or gets more out of the relationship than I do.		1.69	0.712	0.527	−0.898	0.607	0.464	0.770		
15. My partner always wants to know where I am.		2.11	0.821	−0.213	−1.486	0.442	0.286			0.622
Correlation of factors	PRS								0.574	0.653
	IC									0.548

M = media; SD = standard deviation; Skew = skewness; Kurt. = kurtosis; ITC = item-total correlation; ICom = initial communalities; PRS = Perceived Relational Submission; IC = Imposition of the Couple; RRC = Restrictive Relational Control. All correlations between factors present a *p* < 0.001.

**Table 4 behavsci-15-00294-t004:** Descriptive data of the SRPS items and factorial weights in the female sample.

	Subsample 1										Subsample 2			
	M	SD	Skew	Kurt.	ITC	ICom	Fct. Weights				Fct. Weights			
							PRS	IC	RRC	MCSS	PRS	IC	RRC	MCSS
1	1.18	0.445	2.311	4.773	0.546	0.686				0.915				0.896
2	1.17	0.437	2.443	5.461	0.543	0.678				0.926				0.870
3	1.48	0.658	1.036	−0.093	0.549	0.389		0.671				0.754		
4	1.27	0.575	1.994	2.828	0.478	0.326				0.552				0.891
5	1.32	0.617	1.727	1.729	0.496	0.314		0.532				0.692		
6	1.51	0.727	1.054	−0.33	0.535	0.34		0.727				0.593		
7	1.40	0.701	1.446	0.572	0.678	0.539	0.477				0.760			
8	1.30	0.605	1.823	2.084	0.487	0.343			0.680				0.838	
9	1.22	0.543	2.307	4.225	0.689	0.585		0.792				0.782		
10	1.45	0.700	1.234	0.113	0.617	0.507		0.710				0.785		
11	1.70	0.800	0.584	−1.2	0.548	0.423	0.852				0.711			
12	1.32	0.606	1.675	1.633	0.496	0.279	0.504				0.762			
13	1.21	0.519	2.434	4.938	0.686	0.571			0.561				0.887	
14	1.40	0.662	1.395	0.629	0.658	0.547	0.752				0.882			
15	1.83	0.836	0.31	−1.504	0.382	0.216			0.583				0.515	
Correlation of factors														
PRS								0.756	0.457	0.522		0.901	0.757	0.730
IC									0.550	0.575			0.866	0.830
RRC										0.410				0.736

M = mean; SD = standard deviation; Skew = skewness; Kurt. = kurtosis; ITC = item-total correlation; ICom = initial communalities; PRS = Perceived Relational Submission; IC = Imposition of the Couple; RRC = Restrictive Relational Control; MCSS = Male Coercion in Negotiating Safe Sex. All correlations between factors present a *p* < 0.001.

## Data Availability

The data are available on direct request to the corresponding author by e-mail.

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
