# Peer review of "Religiosity, Sexual Double Standard, and Intimate Partner Coercive Victimization in Dating Relationships: An Explanatory Model and Psychometric Evidence"

_behavsci, 2025, doi:10.3390/bs15030294_

Round 1
Reviewer 1 Report
Comments and Suggestions for Authors
This paper presents a study exploring the role of sexual double standard and religiosity in coercive victimization. The paper makes a significant contribution to the field, at the level of analysis it is well executed and the results are adequately discussed. However, there are some aspects that I recommend to be improved. Specifically, the structure and clarity of the introduction, in addition, there is some decision in terms of analysis that requires revision. Also the study presents some clear limitations, since the sample is formed by a young Chilean population (to consider in terms of generalization) and there are relevant variables that have not been considered, such as the duration of the relationship. The study has an eminently psychometric character (EFA, CFA and invariance in two instruments), since examining the structure of two instruments is part of the objectives of the study, I suggest increasing the relevance, using terminology according to international standards (e.g., evidence of validity based on internal structure, evidence of validity based on scale content…). Please see American Educational Research Association, American Psychological Association, & National Council on Measurement in Education. (2014). Standards for educational and psychological testing. AERA Publications Sales. I would also like to share a reflection: although statistically it is possible to use terms of probabilistic causality, an aspect explained by the authors (in lines 154-157), the use of causal language consistently in the manuscript, in my opinión, can be confusing and mislead the reader to erroneous interpretations. I wonder if it would not be advisable to refer to relationships and associations, or, in limitations, to refer explicitly to this issue in a slightly more extended way.
These and other extended comments are presented below, with the intention that they may be useful in the revision of the manuscript.
- - The title may be confusing, it refers to the examination of SDS and religiosity in the context of dating relationships, however, 98% of the sample is single, I suggest reviewing this issue.
- - In the introduction, the information does not follow a common thread that leads the reader through the different variables relevant in the context of IPV. On the other hand, an integrative framework that contextualizes more clearly the role of sexual double standard and religiosity is missing.
- -The introduction also uses causal language (e.g., lines 75, 83...), however, I wonder if the design of these studies allows establishing this type of relationships.
- - In the introduction, different manifestations of IPV are presented. On the other hand, reference is made to coercive control within the couple, however, it gives the impression that this is decontextualized in the model presented previously (lines 44-51 and lines 34-42). I suggest a revision.
- - Similarly, it goes from presentation about coercive control (lines 48-51), and just in the next paragraph it introduces the sexual double standard (SDS) in relation to dating violence and sexual assertiveness. At this point it is not clear what the relationship is between the SDS attitude, dating violence and sexual assertiveness, and why this last variable is mentioned.
- - Reference is made to sex differences in SDS, as a consequence of women's role in gestation and breastfeeding. However, this evidence is partial. No reference is made to the component of social norms, scripts that would underlie this attitude. In terms of clarity, I believe it would be beneficial for the reader to start from the definition of SDS, as a relevant attitude in the context of dating violence, and develop it from there.
- - In lines 61-66, reference is made to the SDS relationship with other psychosexual variables, however, no reference is made to what type of relationship it is found (in a positive, negative sense...).
- - Lines 68-69 actually justify the importance of the sexual double standard in this context, so I suggest the authors expand on this information, and elaborate on it further.
- - In line 70, Chile is referenced for the first time, however it remains decontextualized why the country is mentioned here. Also, lines 89-98 could benefit from a revision in the organization. It starts talking about religion in the population of Chile, then in a study with 11 countries, and finally in latam people from the United States, which is somewhat confusing. I suggest presenting this evidence starting with the most general findings, and ending with the most specific ones, in this case, carried out in Chile, the object of interest of the study.
- - In lines 105-112 reference is made to what is not invariance, however, I wonder if it is so necessary to clarify what it is not, and if it would rather be relevant to focus this section on what it is (from line 113).
- - If the authors have decided to organize the introduction with sections, I suggest separating the objective of the study as well (line 131).
Method
- - In the participants section, I recommend clearly include what the exclusion criteria were. The question has arisen for me as to why up to 29 years of age?
- - In the description of the sample for the 3 objectives, I suggest specifying that the % refers to that of the total sample.
- - I recommend the authors to include validity and reliability evidence of the instruments.
- - Line 203 needs to be referenced.
- - In line 206 reference is made to 10 studies, but these are not referenced.
- - I don’t know if I didn't quite understand, but In lines 211-219, the authors pointed out the use of SRPS, however, I wonder why it has been the authors' choice if it refers to distribution of power within relationships but majority of the sample is single.
- - In the procedure section, it gives the impression that two different ones are mixed: one refers to a pilot test for content validity and the other refers to the main study itself. I suggest separating and clarifying this aspect. Also, the procedure needs to be expanded, what was the data collection period? How was the recruitment of participants done? Was there any incentive to participate?...
- - In data analysis, I recommend the authors to include what is the criterion to consider a level of invariance reached, usually it is a change of less than 0.01 in CFI.
- - In line 290, please indicate again the groups of the multigroup.
- - The sentence in lines 299-302 is confusing and seems incomplete, the authors mean that factor loadings between the groups are not equal? I recommend the authors to review.
- - Why do the authors choose to explore independent models instead of examining whether there is differential item functioning between the groups or items that are not invariant that can be excluded instead of present different models?
- - There are acronyms used for the same CVC, coercive victimization in the couple (CVC) and Cultural Values of Control (CVC), with the latter variable what do the authors refer to?
- - Lines 491-493 need to be referenced.
- - There is no limitation paragraph.
- - The conclusions are extensive and repetitive with the previous information, I suggest presenting a synthesis of the main results.
Other comments;
- In line 568 there is no numbered reference.
- In line 78, there are two [[
- The title of Figure 1 is not in English.
Author Response
Responses to the revisions are included in the attached file. Thank you very much

Reviewer 2 Report
Comments and Suggestions for Authors
See attached document.

Author Response
Responses to the revisions are included in the attached file. Thank you very much.

Round 2
Reviewer 1 Report
Comments and Suggestions for Authors
First of all, I would like to congratulate the authors for their efforts to improve and deepen the study, and for the attention they have paid to my comments.
However, there are still some specific questions that still raise doubts to me:
The authors state that: “We actually explored the relationship variable ‘time in relationship’ as a control variable in the model, but it was eventually dropped. We also explored gender and religious affiliation as control variables in the model. However, these were also dropped. This information is included at the end of the data analysis section.” For greater clarity, I suggest including the results of this analysis.
Causal language is still being used, for example, in lines 208, 446, and 491. I suggest that the authors review this issue throughout the manuscript.
By using separate models, the authors acknowledge the lack of invariance but may overlook potential common structures between groups. Would an approach considering partial invariance provide a more nuanced comparison?
Author Response
Comment 1: The authors state that: “We actually explored the relationship variable ‘time in relationship’ as a control variable in the model, but it was eventually dropped. We also explored gender and religious affiliation as control variables in the model. However, these were also dropped. This information is included at the end of the data analysis section.” For greater clarity, I suggest including the results of this analysis.
Response 1: We have modified the information in the data analysis section, including the mention of the control variables explored and their levels. Additionally, in the results section, we have included the reasons for their exclusion and the corresponding data.
Comment 2: Causal language is still being used, for example, in lines 208, 446, and 491. I suggest that the authors review this issue throughout the manuscript.
Response 2: We have reviewed the examples mentioned; however, there must have been some error, as the indicated lines do not reference the relationship between variables.
In order to improve this point, we have made further changes to the formulation of objectives and hypotheses, as well as to the interpretation of results in the discussion. In the results section (and direct references to results in the discussion), we have maintained the terms direct and indirect effects, as their use is common and methodologically accepted in SEM to describe how latent variables are related within the specified model. Additionally, a clarifying paragraph has been included in the limitations section.
Comment 3: By using separate models, the authors acknowledge the lack of invariance but may overlook potential common structures between groups. Would an approach considering partial invariance provide a more nuanced comparison?
Response 3: We appreciate the reviewer's comment and fully understand the importance of considering potential common structures between groups when addressing factorial invariance. However, we have decided not to apply the partial invariance analysis in this study for several methodological and contextual reasons that we consider crucial for the appropriate interpretation of the results.
First, applying partial invariance entails significant methodological challenges. To properly implement a partial invariance model, rigorous theoretical and empirical criteria must be established to determine which parameters can be considered invariant and which should be freed. This task not only requires a deep understanding of group differences but also a highly detailed approach to avoid misinterpretations. Freeing certain parameters in a partial invariance model can introduce additional complexities that hinder direct comparisons between groups and may lead to conclusions that are not entirely comparable.
Second, one of the instruments analyzed, the RCFS, has a significant limitation: it has not been widely used in men, making it difficult to establish invariant parameters for this group. As a result, applying a partial invariance model is not feasible in this case, given the insufficient data available for men to make a robust and adequate comparison of factorial structures across sexes.
Regarding the DSS, although it has been widely used for both sexes, the available literature on sex differences in its structure is limited, and particularly scarce regarding which parameters might be specific to each sex. While some previous studies suggest certain differences, the lack of consensus and robust evidence on parameter invariance leads us to conclude that, for this analysis, it would be more appropriate to study factorial structures separately. Furthermore, the theoretical differences related to the DSS scale factors between men and women are not yet sufficiently clear in the literature to justify a comparison based on partial invariance.
Nonetheless, we have included a reference to this debate in the manuscript’s limitations section and as a suggestion for future research.
Reviewer 2 Report
Comments and Suggestions for Authors
Thank you for the opportunity to review the revised “Coercive victimization in dating relationships…” manuscript. I appreciate the thoughtful edits and responses to my previous review. This paper is extremely well-written—even more so post-revision--and I am enthusiastic about the work.
Comments:
- I believe the glossary of abbreviations included as an appendix is very helpful and I appreciate its inclusion.
- I am a strong proponent of having very detailed Table titles/captions but will defer to the journal and editors’ preferences on this point.
- I would suggest the authors consult Greenland and Pearce’s article about model building and covariate selection (https://www.annualreviews.org/content/journals/10.1146/annurev-publhealth-031914-122559) and include a more detailed explanation in the methods about their procedure for selecting or rejecting potential covariates (including references that support this methodological choice)--some camps feel very strongly about using statistical significance criteria over theoretical bases for adjustment. The potential for confounding by unmeasured and un-included variables should also be mentioned as a limitation.
Author Response
Comment 1: I am a strong proponent of having very detailed Table titles/captions but will defer to the journal and editors’ preferences on this point.
Response 1: We will maintain the table formatting as per the journal editor’s recommendation.
Comment 2: I would suggest the authors consult Greenland and Pearce’s article about model building and covariate selection (https://www.annualreviews.org/content/journals/10.1146/annurev-publhealth-031914-122559) and include a more detailed explanation in the methods about their procedure for selecting or rejecting potential covariates (including references that support this methodological choice)--some camps feel very strongly about using statistical significance criteria over theoretical bases for adjustment. The potential for confounding by unmeasured and un-included variables should also be mentioned as a limitation.
Response 2: Thank you very much for suggesting this article, as it provides valuable insights into understanding the role of covariances and refining the findings of the present study.
As mentioned in Greenland and Pearce’s study in the section "CURRENT MODELING STRATEGIES FOR CONFOUNDER CONTROL," in the present article, we have attempted to adjust for all theoretically relevant variables rather than following the procedure suggested by the authors. Nevertheless, we have included the reasons why certain control variables were excluded from the model (lines 484–489 and 519–524) and have acknowledged the limitations regarding the variables not included.
Round 3
Reviewer 1 Report
Comments and Suggestions for Authors
I would like to thank the authors for their attention to my comments.